# Real time dynamics of Gating-Related conformational changes in CorA

Martina Rangl[1,2], Nicolaus Schmandt[3], Eduardo Perozo[3]*, Simon Scheuring[1,2]*

[1]Department of Anesthesiology, Weill Cornell Medical College, New York, United States; [2]Department of Physiology and Biophysics, Weill Cornell Medical College, New York, United States; [3]Department of Biochemistry and Molecular Biophysics, The University of Chicago, Chicago, United States

**Abstract** CorA, a divalent-selective channel in the metal ion transport superfamily, is the major $Mg^{2+}$-influx pathway in prokaryotes. CorA structures in closed ($Mg^{2+}$-bound), and open ($Mg^{2+}$-free) states, together with functional data showed that $Mg^{2+}$-influx inhibits further $Mg^{2+}$-uptake completing a regulatory feedback loop. While the closed state structure is a symmetric pentamer, the open state displayed unexpected asymmetric architectures. Using high-speed atomic force microscopy (HS-AFM), we explored the $Mg^{2+}$-dependent gating transition of single CorA channels: HS-AFM movies during $Mg^{2+}$-depletion experiments revealed the channel's transition from a stable $Mg^{2+}$-bound state over a highly mobile and dynamic state with fluctuating subunits to asymmetric structures with varying degree of protrusion heights from the membrane. Our data shows that at $Mg^{2+}$-concentration below $K_d$, CorA adopts a dynamic (putatively open) state of multiple conformations that imply structural rearrangements through hinge-bending in TM1. We discuss how these structural dynamics define the functional behavior of this ligand-dependent channel.

*For correspondence:
eperozo@uchicago.edu (EP);
sis2019@med.cornell.edu (SS)

**Competing interests:** The authors declare that no competing interests exist.

## Introduction

Magnesium ($Mg^{2+}$) is a key divalent cation in biology. *bala*It regulates and maintains numerous, physiological functions such as nucleic acid stability, muscle contraction, heart rate and vascular tone, neurotransmitter release, and serves as cofactor in a myriad of enzymatic reactions (*Altura, 1991*; *de Baaij et al., 2015*; *Jahnen-Dechent and Ketteler, 2012*; *Romani, 2013*; *Ryan, 1991*). Most importantly, it coordinates with ATP, and is thus crucial for energy production in mitochondria (*Altura, 1991*; *Pilchova et al., 2017*; *Romani, 2011*; *Swaminathan, 2003*; *Yamanaka et al., 2016*). In order to store $Mg^{2+}$ in the mitochondrial lumen it is imported via Mrs2 (*Kolisek et al., 2003*) and Alr2 (*Liu et al., 2002*) ion channels that are closely related to CorA, the main $Mg^{2+}$-importer in bacteria (*Guskov and Eshaghi, 2012*; *Hmiel et al., 1986*; *Knoop et al., 2005*; *Schweyen and Froschauer, 2007*). Although these $Mg^{2+}$-transport proteins do not show much sequence conservation, they all share two trans-membrane domains (TMDs) with the signature motif Glycine-Methionine-Asparagine (GMN) at the extracellular loop (*Knoop et al., 2005*; *Schweyen and Froschauer, 2007*).

The crystal structure of CorA from *Thermotoga maritima* in its $Mg^{2+}$-bound closed state (*Eshaghi et al., 2006*; *Lunin et al., 2006*; *Payandeh and Pai, 2006*) revealed a 5-fold symmetric homo-pentamer forming an ~11 nm funnel-like structure, with large intracellular domains and $Mg^{2+}$-ions bound between the subunits (*Eshaghi et al., 2006*; *Guskov et al., 2012*; *Lerche et al., 2017*; *Lunin et al., 2006*; *Payandeh and Pai, 2006*). Through CorA, $Mg^{2+}$-homeostasis is achieved by a negative feedback mechanism in which $Mg^{2+}$ acts as both, charge carrier and ligand (*Dalmas et al., 2014b*). The binding of $Mg^{2+}$ at the cytoplasmic subunit interfaces leads to channel closing and thereby limits further $Mg^{2+}$-influx (*Dalmas et al., 2014a*; *Dalmas et al., 2014b*; *Palombo et al., 2012*; *Pfoh et al., 2012*; *Schindl et al., 2007*).

Determining the structure of CorA in its $Mg^{2+}$-unbound (Apo) form is considered a fundamental step towards understanding its gating mechanism. Yet, the first CorA crystal structure in the absence of divalent cations showed little or no changes compared to the fully $Mg^{2+}$-bound closed channel structure, with only a slight kink between TMDs and intracellular domains (*Pfoh et al., 2012*). In contrast, EPR spectroscopy indicated much larger structural rearrangements within the protein upon $Mg^{2+}$-dissociation, suggestive of a more dramatic closed-to-open channel transition (*Dalmas et al., 2010*; *Dalmas et al., 2014b*). These results were confirmed by cryo-electron microscopy (cryo-EM), reporting surprising asymmetric rearrangements of the individual subunits within the homo-pentamer in $Mg^{2+}$-free condition (*Cleverley et al., 2015*; *Matthies et al., 2016*). Based on ~7 Å cryo-EM structures (*Matthies et al., 2016*), a model was proposed in which $Mg^{2+}$-unbinding leads to an increase in the inter-subunit conformational flexibility of CorA, thereby resulting in at least two asymmetric structures with, presumably, open gates (*Matthies et al., 2016*).

Despite progresses in the structural characterization of the $Mg^{2+}$-free CorA structure(s), the transition from the closed ($Mg^{2+}$-bound) state to the conductive unliganded conformation is still unclear. At 7.1 Å, the putative open CorA asymmetric cryo-EM structures were solved only from a relatively low number of particles (26,271 and 27,416 of 173,653 particles for open-I and open-II states, respectively) (*Matthies et al., 2016*), and due to its intrinsic averaging for structure determination, the cryo-EM data cannot inform about time-dependent behavior. We posit that these structures could be sampled from three different dynamic processes: they could represent two different open state conformations with significant state dwell-times (structurally resolved by cryo-EM); they could be intermediates along a conformational trajectory; or finally, they might be mere snapshots of a highly dynamic, fluctuating molecule without a long-lasting and well-defined high-resolution structure.

In order to elucidate the sequence of events underlying gating-related CorA conformational changes, we used high-speed atomic force microscopy (HS-AFM) (*Ando et al., 2001*; *Ando et al., 2014*) to capture CorA structural rearrangements during $Mg^{2+}$-depletion experiments. HS-AFM is unique in its ability to concomitantly characterize molecular structures and dynamics under native-like conditions. In agreement with electrophysiological, spectroscopic and biophysical data (*Dalmas et al., 2014b*), we found that during $Mg^{2+}$-depletion experiments individual CorA molecules lost pentameric symmetry and became highly dynamic at $Mg^{2+}$-concentrations below ~2 mM. Under these conditions, the symmetric state is reversibly adopted by means of dynamic fluctuations among the individual subunits, indicative of spontaneous $Mg^{2+}$-rebinding. However, in the absence of $Mg^{2+}$, CorA transitions to an asymmetric state. Thus, the conformational energy landscape likely comprises two deep energy minima represented by the fully symmetric ($Mg^{2+}$-bound) and the asymmetric ($Mg^{2+}$-depleted) states interspersed by a wide plateau of conformational fluctuations of a highly flexible molecule, which likely represents the conductive state.

## Results

### Lipid composition determines reconstitution density and morphology

In recent years, HS-AFM has demonstrated its power to observe molecular mechanisms and structural dynamics of single molecules under physiological conditions with extraordinary detail (*Chiaruttini et al., 2015*; *Kodera et al., 2010*; *Marchesi et al., 2018*; *Preiner et al., 2014*; *Rangl et al., 2016*; *Ruan et al., 2018*; *Ruan et al., 2017*). Here, we have used HS-AFM to track the conformational changes of CorA as a result of $Mg^{2+}$-concentration changes in real-time. For this, wild-type (wt) CorA was reconstituted at low lipid-to-protein ratios (LPRs between 0.2 and 0.4) in the presence of $Mg^{2+}$. The resulting proteo-liposomes were adsorbed onto freshly cleaved mica under saturating $Mg^{2+}$ (10 mM) condition and imaged at an acquisition rate of 1–2 frames $s^{-1}$ with a resolution of 0.5 nm $pixel^{-1}$. CorA-containing vesicles were generated from protein reconstitutions in POPC/POPG (3:1, w:w). When imaged by HS-AFM, vesicles spread on the mica support resulting in large membrane patches with densely packed CorA (*Video 1*). These patches mostly exposed the periplasmic face. Smaller crowded areas of molecules expose the intracellular side (*Figure 1a*, *Video 1*).

Cross-section analysis of the CorA membrane patches showed overall heights of ~12 nm (*Figure 1b*). High-resolution movies of the periplasmic domains revealed protrusions of 0.9 ± 0.4 nm

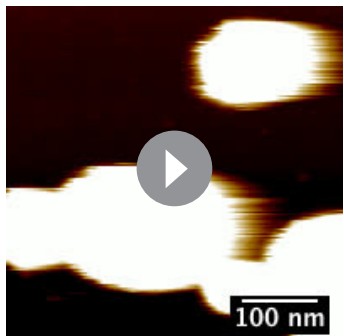

**Video 1.** CorA reconstituted in POPC/POPG liposomes. In overview scans the proteoliposomes were opened by applying slightly increased loading forces, thereby revealing CorA membranes exposing both faces, the periplasmic face in crystalline packing and the intracellular side in crowded membrane areas. Video settings: Full scan size: 444 nm, 200 pixels, scan rate: 1 frame s$^{-1}$, full color range: 20 nm.
https://elifesciences.org/articles/47322#video1

in height and 3.4 ± 0.4 nm in diameter, and resolved a central indentation where the channel pore is located (*Figure 1c*). Despite the fast diffusion of CorA molecules within the clusters exposing the intracellular side, some high-resolution snapshots allowed analysis of the surface structure. The intracellular face protruded ~7 nm from the membrane (*Figure 1—figure supplement 1a*) with top-ring and outer diameters of ~6 nm and ~10 nm, respectively (*Figure 1—figure supplement 1b,c*; *Video 2*). These measurements are in good agreement with the molecular dimensions of CorA (*Eshaghi et al., 2006*; *Lunin et al., 2006*; *Matthies et al., 2016*).

However, it became clear that in order to study Mg$^{2+}$-dependent conformational dynamics, more stably packed molecules were required (*Müller et al., 1999*; *Ramadurai et al., 2010*). We found that reconstituting the protein in DOPC/DOPE/DOPS (4:5:1, w:w:w) and adsorbed and imaged at lower pH 6.0 resulted in widespread surface coverage of CorA-crowded membranes with only slowly moving molecules.

Moreover, under these conditions densely packed CorA patches were stacked and exposed the intracellular face of the channel (*Figure 1d,e*; *Video 3*), thus providing an excellent experimental platform for studying CorA Mg$^{2+}$-dependent conformational changes. High-resolution HS-AFM topographs displayed a 'flower-shaped' surface structure corresponding to the CorA intracellular face. This view allowed to resolve individual subunits in the pentamer with top-ring diameter of 5.0 ± 0.9 nm and a center-to-center distance of the molecules, that is outer diameter, of 10.9 ± 2.1 nm (*Figure 1f*).

## Mg$^{2+}$-depletion induces large conformational changes of the intracellular face

We then monitored the structural changes of the channel upon Mg$^{2+}$-depletion in real time at higher magnification, that is at scan sizes < 200 nm and pixel sampling of 0.5 nm pixel$^{-1}$. First, we studied the periplasmic face of CorA under saturating Mg$^{2+}$ (10 mM), in which all channels are expected to be in the closed conformation. Subsequently, membranes were imaged at reduced Mg$^{2+}$-concentration near the reported apparent $K_d$ for Mg$^{2+}$ (~2 mM), and finally at 0 mM Mg$^{2+}$ to focus on the dynamically open Apo form.

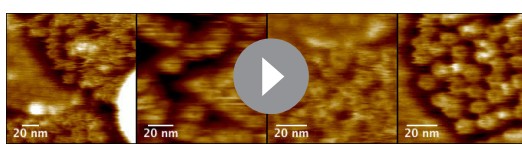

**Video 2.** Four examples of high magnification movies of the intracellular side of CorA reconstituted in POPC/POPG. Fast diffusing molecules with flower-shaped structures were observed. Video settings: Full scan sizes: 80–120 nm, 160–200 pixels, scan rates: 1–2 frame s$^{-1}$, full color range: 5 nm.
https://elifesciences.org/articles/47322#video2

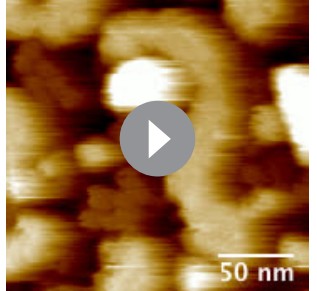

**Video 3.** CorA reconstituted in DOPC/DOPE/DOPS 4:5:1. The sample support was fully covered with CorA membranes exposing the intracellular face with areas of slowly moving molecules and membranes with densely packed channels stacked on top. Video settings: Full scan size: 200 nm, 200 pixels, scan rate: 1 frame s$^{-1}$, full color range: 20 nm.
https://elifesciences.org/articles/47322#video3

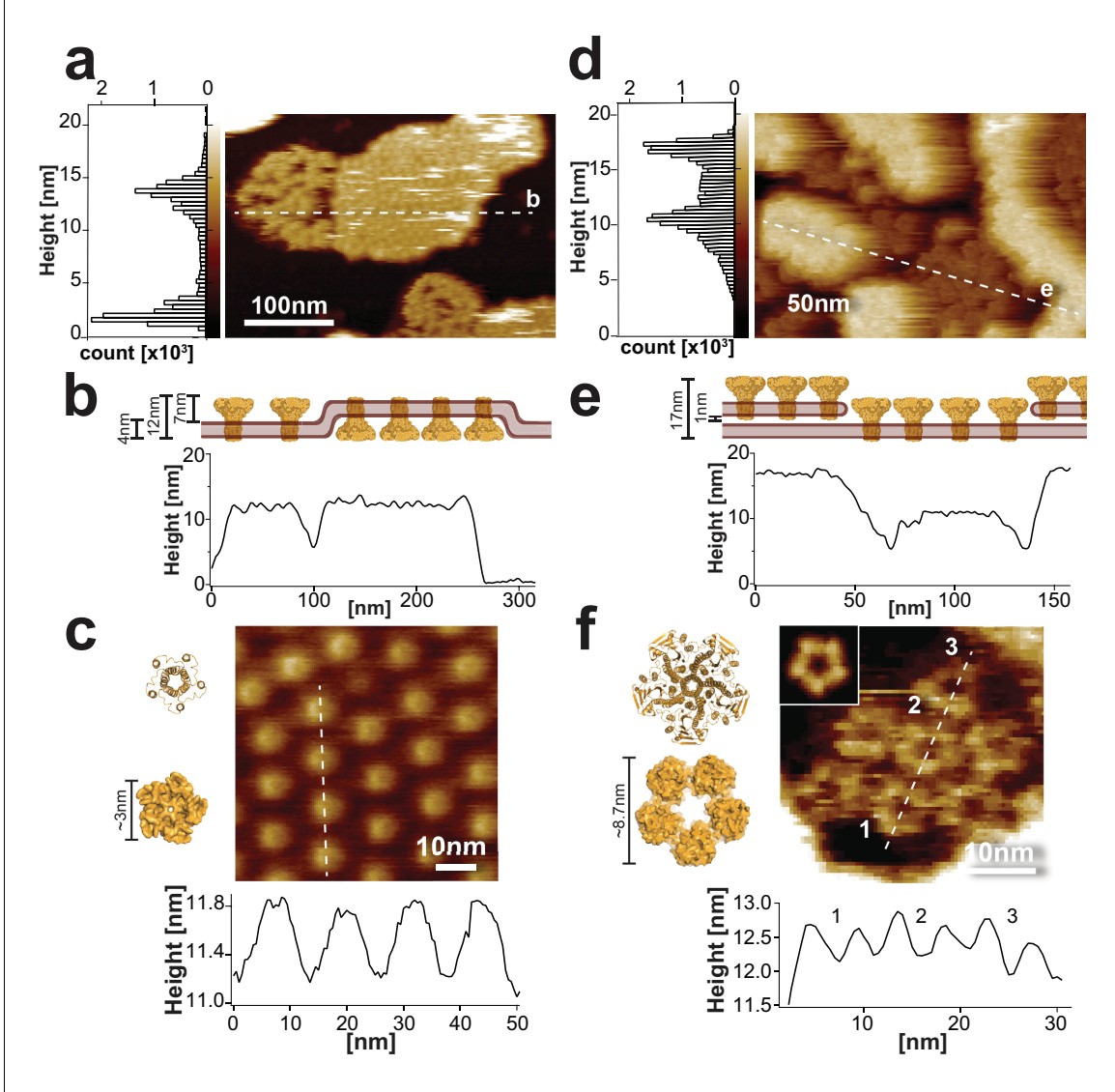

**Figure 1.** Sample morphology of CorA reconstitutions for HS-AFM. (a) HS-AFM overview topograph of densely packed CorA in a POPC/POPG (3:1) lipid bilayer exposing the periplasmic side and a loosely packed protein area with diffusing molecules exposing the intracellular face (full color scale: 20 nm). Left: Height histogram of the HS-AFM image with two peaks representative of the mica and the CorA surface (ΔHeight (peak-peak): 12 nm (20,500 height values)). The dashed line indicates the position of the cross-section analysis shown in (b). (b) Profile of the membrane shown in a), including a cartoon (top) of the membrane in side view. The height profile (~12 nm) corresponds well to the all-image height analysis (a, left) and the CorA structure (*Matthies et al., 2016*). (c) High-resolution image (top) and cross-section analysis along dashed line (bottom) of the periplasmic face. The height and dimension of the periplasmic face is in good agreement with the structure (left), and the periodicity (~14 nm, n = 40) corresponds well with the diameter of the intracellular face spacing the molecules on the other side of the membrane (full color scale: 2 nm). (d) HS-AFM image of densely packed CorA embedded in a DOPC/DOPE/DOPS (4:5:1) membrane. This reconstitution resulted in two stacked membrane layers, both exposing the CorA intracellular face. The dashed line indicates the position of the cross-section analysis shown in (e). Left: Height histogram of the HS-AFM image with two peaks at ~12 nm and ~17 nm (32,500 height values), corresponding to the proteins in two stacked membranes (full color scale: 20 nm). (e) Section profile of the membrane shown in d), including a cartoon (top) of the membrane in side view. (f) High-resolution view and cross-section analysis along dashed line (bottom) of the CorA intracellular face revealing the individual subunits of the pentamers (full color scale: 3 nm). Inset: 5-fold symmetrized average of CorA. The dimensions of CorA observed with HS-AFM are in good agreement with the structure (left: PDB 3JCF). The structures in (c) and (f) are shown in ribbon (top) and surface (bottom) representations, respectively.

The online version of this article includes the following figure supplement(s) for figure 1:

**Figure supplement 1.** Characterization of CorA intracellular side reconstituted in POPC/POPG membranes.

$Mg^{2+}$-depletion was achieved by live injection of EDTA, while continuously monitoring CorA. Using the closed state as reference, no structural changes of the periplasmic face of CorA were observed at 10 mM $Mg^{2+}$, at 2 mM $Mg^{2+}$ (after ~10 min) and in absence of $Mg^{2+}$ (after ~12 min) (*Figure 2—figure supplement 1*). At the present resolution (0.5 nm pixel$^{-1}$) any putative changes in the conformation of the periplasmic face of the channel are beyond the current resolution limit of HS-AFM (although we cannot rule out that the structures of the small periplasmic loops might not change upon $Mg^{2+}$-depletion). This is consistent with the ~7 Å cryo-EM structures, which show little or no $Mg^{2+}$-dependent changes on the periplasmic face (*Matthies et al., 2016*; *Pfoh et al., 2012*).

In stark contrast, previous EPR (*Dalmas et al., 2010*; *Dalmas et al., 2014b*) and cryo-EM (*Matthies et al., 2016*) analyses have revealed dramatic conformational rearrangements of the intracellular domains (resulting in a loss of symmetry) in the nominal absence of $Mg^{2+}$ (conditions favoring the functionally open state) (*Figure 2a*). First, to measure the CorA $Mg^{2+}$-affinity (*Pilchova et al., 2017*) in our experimental set-up, we used a microfluidic system connected to a constant pressure and flow pump (*Miyagi et al., 2016*), with which we slowly exchanged the complete 10 mM $Mg^{2+}$ measuring solution to a $Mg^{2+}$-free buffer (containing additional 2 mM EDTA). Analysis of the distribution of symmetric and dynamically asymmetric CorA particles pointed to a $K_d$ of ~2 mM $Mg^{2+}$, which is in good agreement with the reported affinity (*Figure 2—figure supplement 2*). Next, we monitored the structural and dynamical transition of CorA upon $Mg^{2+}$-depletion by pipetting defined amounts of EDTA into the measurement fluid cell to achieve the following equilibrium concentrations: 10 mM $Mg^{2+}$, ~2 mM $Mg^{2+}$ (~$K_d$), and 0 mM $Mg^{2+}$ (putative).

Consistent with the $Mg^{2+}$-liganded cryo-EM structure (*Figure 2a*, left), at saturating 10 mM $Mg^{2+}$ condition CorA channels revealed a stable, flower-like 5-fold symmetric structure (*Figure 2b*, left). However, once $Mg^{2+}$-concentrations dropped to ~2 mM and below, individual channels start to fluctuate between various structural states, occasionally assuming increased height and, after ~20 min in absence of $Mg^{2+}$, adopting an ill-defined conformation with significantly increased protrusion height (*Figure 2b*, right). The $Mg^{2+}$-dependent loss of symmetry and increased protrusion height are also consistent with expected structural features of the $Mg^{2+}$-free (open) CorA structures, where individual subunits move towards the former 5-fold axis and thus stand taller (*Matthies et al., 2016*) (*Figure 2a*, bottom). The large, ~1.5 nm, protrusion height difference (ΔHeight) of such a conformation change presents a useful signature to detect and follow the conformational dynamics of individual channels in the membrane. However, these two ΔHeight-states must not be mistaken with functional or even structural states. They are merely a way to topographically discriminate between the closed 5-fold symmetric state and any other state where single subunits stand up, that is move towards the 5-fold axis and thus appear higher.

Height section kymographs of individual molecules (*Figure 3a*, top) over extended imaging periods (*Figure 3a*, middle) allowed the subsequent computational detection of single molecule dynamics based on the fluctuations in height differences (ΔHeight) as a function of time (*Figure 3a*, bottom): While a ΔHeight of ~1 nm represents a symmetric CorA structure with a pore in the center, larger ΔHeight values represent molecules with increased protrusion height. ΔHeight values < 1 nm were found for molecules of low height (putative closed state) where the central pore could not be resolved. Hence, the ΔHeight/time traces were idealized to represent two ΔHeight-states using the Step Transition and State Identification (STaSI) algorithm developed for single molecule experiments (*Shuang et al., 2014*), which we successfully adapted for the analysis of HS-AFM ΔHeight/time traces (*Heath and Scheuring, 2018*) (*Figure 3a*, red line).

We monitored molecular transitions from the $Mg^{2+}$-bound to $Mg^{2+}$-free states in time-lapse experiments. HS-AFM images of CorA membrane patches during these extended experiments reproducibly revealed the conformational changes of individual molecules over time (*Figure 3b*, *Video 4*, *Video 5*). While CorA maintained the 5-fold symmetric state at saturating $Mg^{2+}$ condition (*Figure 3b*, green time stamps), the channels started switching dynamically between (at least) two conformational states at concentrations below ~2 mM $Mg^{2+}$ (*Figure 3b*, blue and early red time stamps), and began populating strongly protruding asymmetric states as $Mg^{2+}$-depletion progressed (*Figure 3b*, red time stamps). A cumulative height-based state-assignment of CorA as a function of $Mg^{2+}$-depletion demonstrated that the number of channels in the putatively open state (s) gradually increased (*Figure 3b*, bottom). In contrast, the number of transitions (*i.e.* switching back and forth between the high (open) state(s) and the state of lower height) cumulated earlier during the titration experiment (*Figure 3c*). Thus, we are looking at a transition from the symmetric low

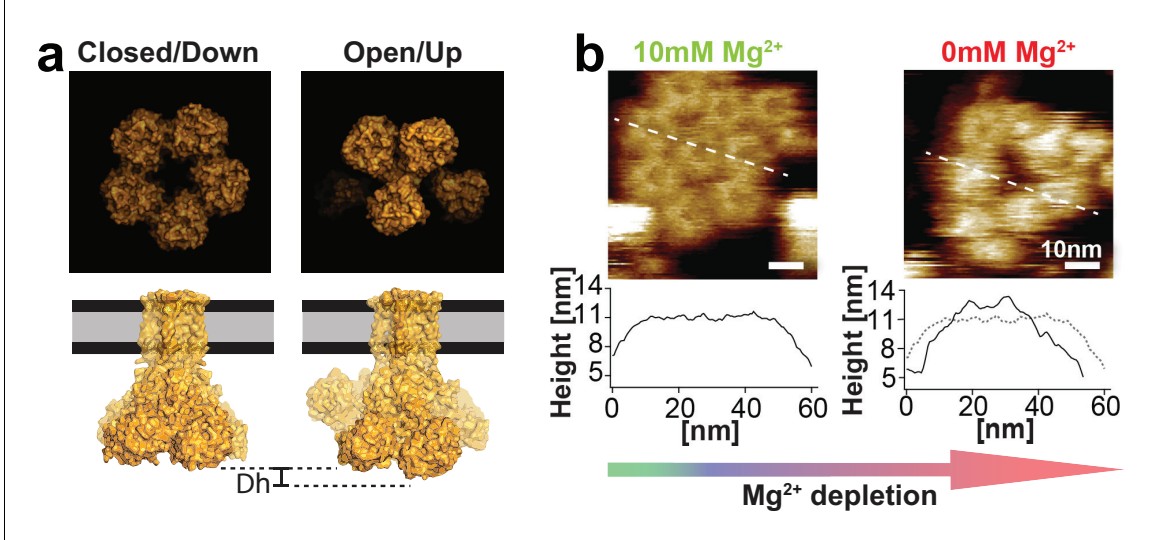

**Figure 2.** CorA in presence and absence of Mg$^{2+}$. (a) Molecular surface representations of CorA structures in the Mg$^{2+}$-bound (closed) and Mg$^{2+}$-free (open) states (PDB: 3JCF and 3JCH) (*Matthies et al., 2016*). Top: intracellular view. Bottom: side view. The Mg$^{2+}$-free (open) structure protrudes further (Δh) from the membrane. (b) HS-AFM images of a membrane patch with densely packed CorA exposing the intracellular face with corresponding cross-section analyses in 10 mM Mg$^{2+}$ (left) and after ~20 min in absence of Mg$^{2+}$ (right). The cross-section profiles (bottom) along the dashed lines demonstrate the height increase of ~1.5 nm of the same molecules in absence of Mg$^{2+}$ compared to the topography height in presence of Mg$^{2+}$. The online version of this article includes the following figure supplement(s) for figure 2:

**Figure supplement 1.** Monitoring the CorA periplasmic side upon Mg$^{2+}$-depletion.

**Figure supplement 2.** Controlled buffer exchange experiment confirming the intracellular CorA Mg$^{2+}$-sensor domain affinity for Mg$^{2+}$.

(closed) to the asymmetric high (open) structures and a wide range of fluctuations between states at intermediate Mg$^{2+}$-concentrations, or within the first minutes of complete Mg$^{2+}$-depletion.

## CorA state transition revealed a highly dynamic intermediate

In addition to the gradual state conversion during Mg$^{2+}$-depletion, the number of transition events varied over time. At saturating Mg$^{2+}$, all molecules occupied a stable conformation and the state interconversion activity is virtually zero (*Figure 3b*, green). About 5–8 min into the complete Mg$^{2+}$-depletion experiment (at 0 mM Mg$^{2+}$) the dynamics of the channels reach a maximum (*Figure 3c*), followed by another more stable, strongly protruding and asymmetric state, putatively representing an open conformation (*Figure 3b,c*, red). The

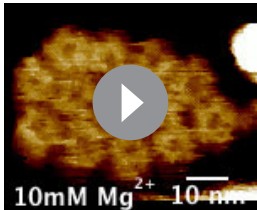

**Video 4.** CorA conformational changes upon Mg$^{2+}$-depletion. Within ~15 min most molecules transition from a 5-fold symmetric, Mg$^{2+}$-bound, putatively closed, conformation to dynamically active, flexible structures that equilibrate into an asymmetric molecule of increased height, the putative ligand-free open state. Video settings: Full scan size: 60 × 40 nm, 0.4 nm/pixel, scan rate: 1.3 frame s$^{-1}$, full color range: 5 nm.
https://elifesciences.org/articles/47322#video4

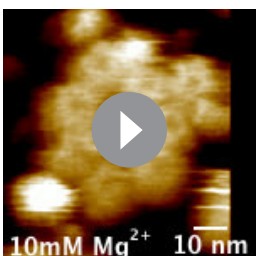

**Video 5.** Monitoring the conformational changes upon Mg$^{2+}$-depletion. Within ~30 min, CorA molecules change from the stable 5-fold symmetric, putatively closed, conformation to a dynamic structure representing the apparent open conformation. Video settings: Full scan size: 100 nm, 200 pixels, scan rate: 1 frame s$^{-1}$, full color range: 5 nm.
https://elifesciences.org/articles/47322#video5

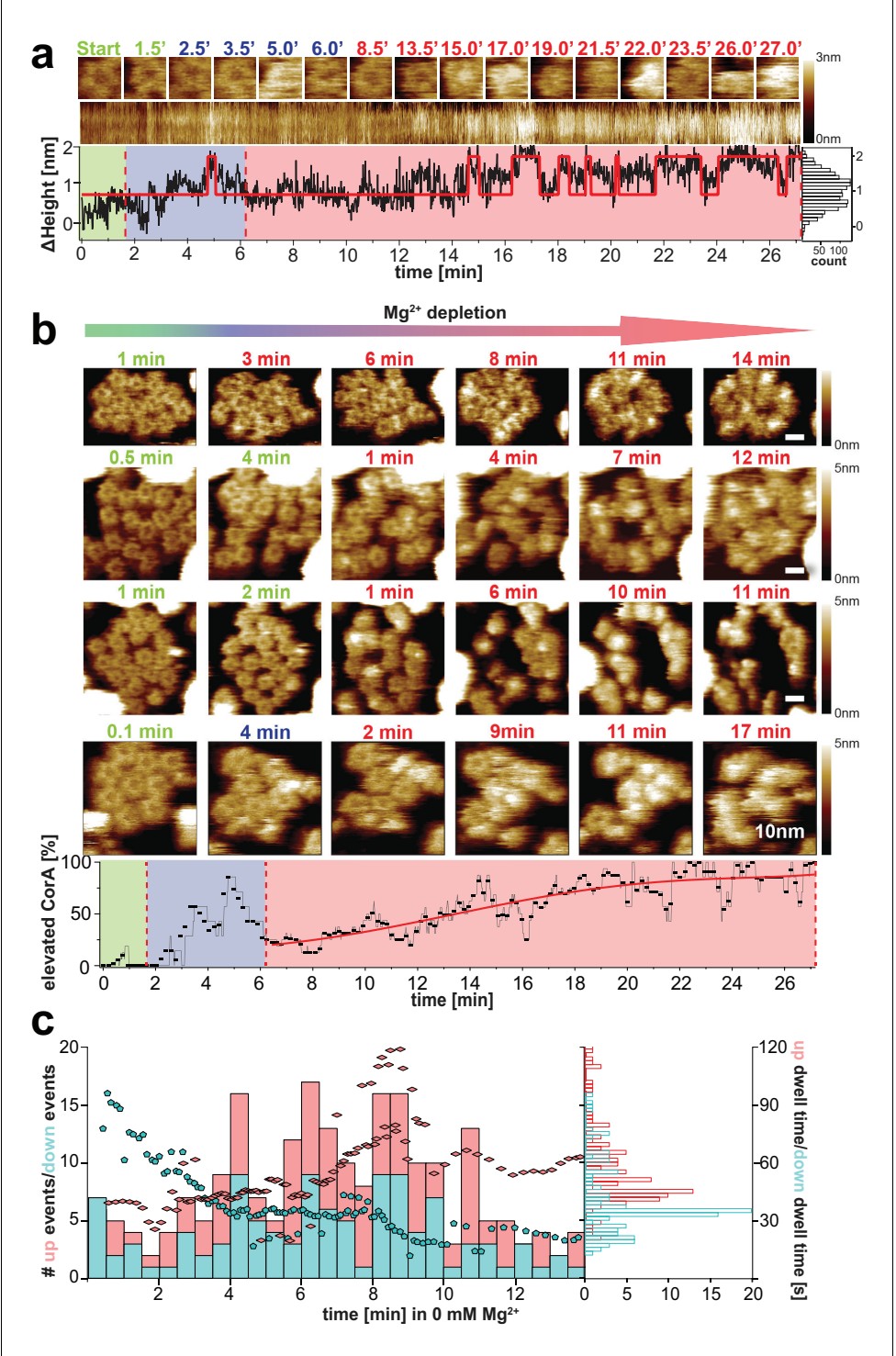

**Figure 3.** CorA conformational changes and dynamics upon Mg$^{2+}$-depletion. (a) Single CorA molecule at indicated time points during Mg$^{2+}$-depletion. Below: Section kymograph of the molecule and corresponding ΔHeight/time trace derived from the center area of the imaged CorA channel. The red line is a fitted idealized trace with two distinct ΔHeight-states. Right: Height histogram of the ΔHeight/time trace. (b) Time-lapse HS-AFM of membrane patches with densely packed CorA channels that expose the intracellular face during Mg$^{2+}$-depletion experiments. Direction of Mg$^{2+}$-depletion and time points are indicated above frames. Scale bars: 10 nm. Below: Percentage of CorA molecules with increased height (putatively open states) as a function of time. (a) and (b): Frames acquired in saturating 10 mM Mg$^{2+}$-concentrations are indicated in green, at ~2 mM Mg$^{2+}$ in blue (only tested in (a) and the bottom panel of (b)) and at complete Mg$^{2+}$-depletion (0 mM Mg$^{2+}$) in red. Depletion of Mg$^{2+}$ was achieved by manual addition of EDTA. (c) Number of Δheight-transitions and associated dwell-times following Mg$^{2+}$-depletion. Bars indicate the number of high-to-low ('down', turquois) and

*Figure 3 continued on next page*

*Figure 3 continued*

low-to-high ('up', red) events binned over a time window of 30 s. Turquois pentagons and red diamonds indicate average closing and opening event time-points of corresponding average dwell-time (right axis), respectively. These averages were calculated over a sliding window of 20 events along the time axis. Analysis included molecules from 2 experiments at 0 mM $Mg^{2+}$ (shown in (b) top and bottom) and 30,500 HS-AFM images thereof. Right: Histogram of average dwell-times in the 'high' (red) and 'low' (turquois) states (where the high state represents/comprises all conformational states with elevated subunits).

time windows of these observations correlate well with electrophysiology experiments, in which the CorA-driven $Mg^{2+}$-currents in *Xenopus leavis* oocytes decay within 15–20 min (*Dalmas et al., 2010*; *Dalmas et al., 2014b*). In addition to the number of transitions, the same trend is represented when analyzing the dwell-times of the ΔHeight-states in the nominal absence of $Mg^{2+}$ (*Figure 3c*): Within the first 5 min in 0 mM $Mg^{2+}$ the average time spent in the closed (low height) state is almost twice (~60 s) the averaged dwell-time in one of the putatively open (elevated height) conformations. After ~10 min in $Mg^{2+}$-free condition, the situation is reversed, and more molecules show long dwell-times in the elevated state (*Figure 3c*, turquois pentagons and red diamonds). Thus, in this experiment, the CorA gating pathway initially favors the stable $Mg^{2+}$-bound, low-height state by ~-$0.7k_BT$ while elevated $Mg^{2+}$-free (open) conformation(s) are favored when $Mg^{2+}$ is depleted from all binding sites. In between, the channel is in a highly dynamic regime, probably reflecting $Mg^{2+}$-unbinding and -rebinding events (*Figure 3*, *Videos 4* and *5*). We must highlight, however, that a total depletion of $Mg^{2+}$ probably never occurs under physiological conditions, thus the fully $Mg^{2+}$-depleted asymmetric stable states observed by cryo-EM and likely adopted here at the end of the depletion experiments might not be visited in the cell. More likely, the 5-fold symmetric closed state interchanges with a highly fluctuating molecule where single subunits dissociate from the quaternary structure of the cytoplasmic ensemble, opening the channel.

## CorA fluctuates between several conformations in the open state

Following the general observation that the stable closed and open conformations are interconnected by a regime in which the channels are highly dynamic, we pursued a detailed structural examination of CorA in this intermediate dynamic stage at low $Mg^{2+}$-concentrations. High-resolution HS-AFM image sequences of individual CorA channels revealed that CorA undergoes conformational rearrangements of the entire cytosolic $Mg^{2+}$-sensor domain at rates beyond the bandwidth of the current measurements (<550 ms, the imaging rate of our videos). We also acquired movies at 250 ms frame acquisition and found that individual sequential frames displayed different apparently unrelated conformations, an indication that the conformational fluctuations are even faster (*Figure 4d*).

Movie snapshots were classified into four conformational classes (*Figure 4*, *Video 6*): First, the fully-liganded 5-fold symmetric closed conformation (*Eshaghi et al., 2006*; *Lunin et al., 2006*; *Matthies et al., 2016*; *Payandeh and Pai, 2006*) (*Figure 4a*, top row, *Figure 4b*, 3.85 s, 7.15 s, 8,8 s and 14.3 s and *Figure 4c*, 1.1 s, 5.5 s and 6.05 s, *Figures 4d*, 0s, 0.75 s and 2.25 s). Second, a structure of reduced diameter with three asymmetrically distributed protrusions of increased height (*Figure 4a*, second row, *Figure 4b*, 9.35 s, *Figure 4c*, 0.55 s, 4.95 s, *Figure 4d*, 0.5 s), likely corresponding to the cryo-EM open-I state (*Matthies et al., 2016*). Third, a round- or star-shaped conformation with an elevated center resembling the cryo-EM open-II state where one subunit is displaced towards the channel axis (*Matthies et al., 2016*) (*Figure 4a*, third row, *Figure 4b*, 0s, 6.05 s, 7.7 s, 17.05 s and *Figures 4c*, 0s, 0.55 s, 2.75 s). However, given that our tentative assignment to open-I and open-II states only covers a fraction of all molecular observations, a fourth category was invoked to include the remaining range of structural variations, which we named open-+ states. This category includes asymmetric molecules of all kinds, even those without increased height (*Figure 4a*, fourth row, *Figure 4b*, 16.5 s, *Figure 4d*, 1.5 s, 1.75 s). It is clear that a key challenge in defining the conformational landscape of CorA will be the unbiased classification of discrete states in this highly flexible molecule, made evident by following a single CorA channel frame by frame (*Figure 4c,d*). Unfortunately, all our computational classification attempts failed likely due to the presence of protein-protein contacts with neighboring molecules and limited resolution due to the high mobility.

To work out a more detailed picture of the CorA-transitions we visually assigned these four different types of states to high-resolution HS-AFM movie sections recorded at different $Mg^{2+}$-

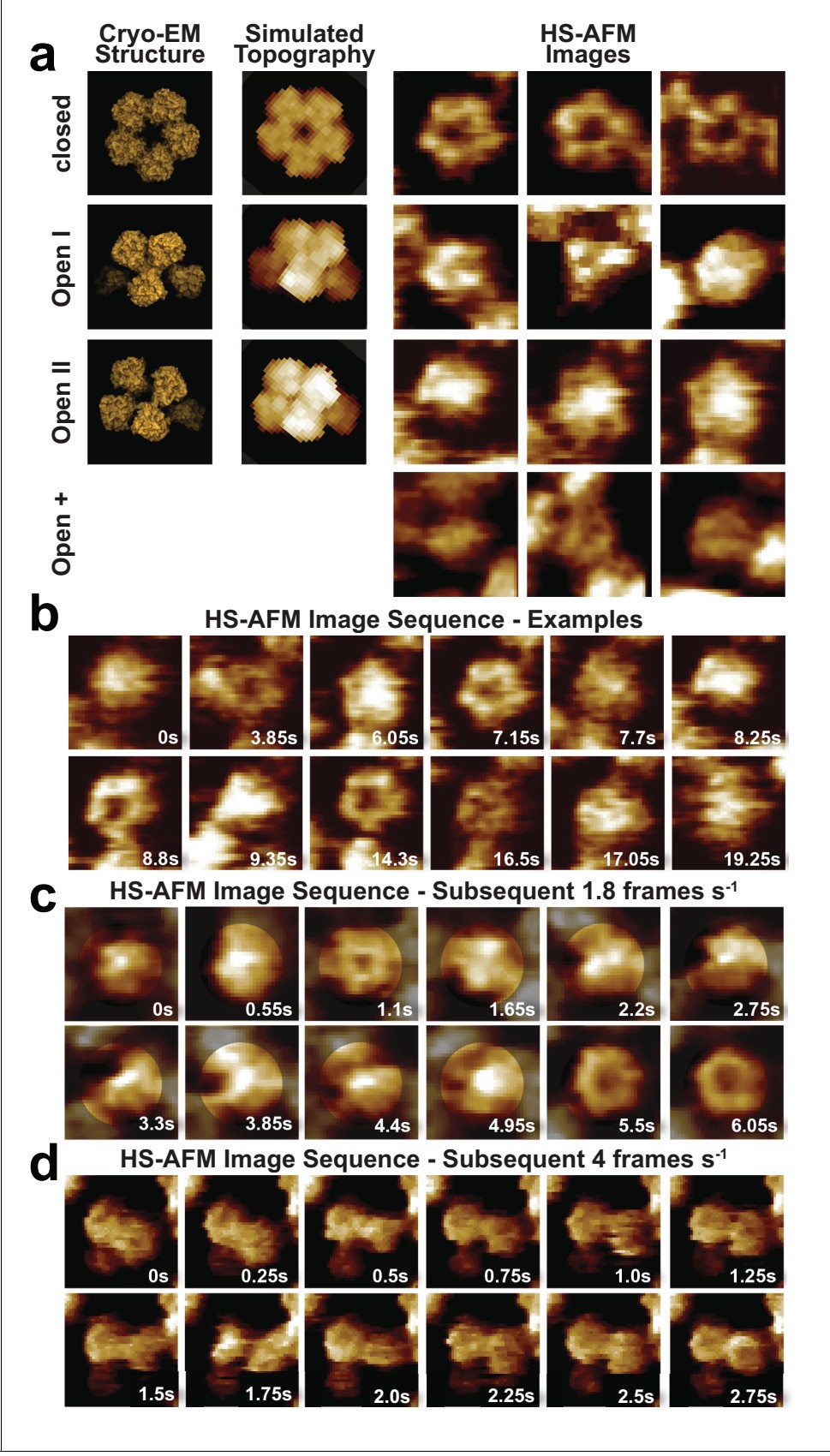

**Figure 4.** CorA adopts several highly dynamic conformations. (a) Left: Surface representations of CorA cryo-EM structures in the high-resolution closed (PDB: 3JCF) and the two 7 Å resolution $Mg^{2+}$-free open (PDBs 3JCH, 3JCG) conformations. Center: AFM topography simulations of the structures on the left. Right: Examples of HS-AFM frames of single CorA molecules in the symmetric closed (upper panel) and the asymmetric (putatively open) conformations (bottom panels). Full-frame size: 17.5 nm. Full z-scale: 2 nm. (b) High-resolution HS-AFM frames of an individual CorA channel upon $Mg^{2+}$-depletion. The molecule fluctuates dynamically between several conformations. Time stamps are indicated. Full-frame size: 17.5 nm. Full z-scale: 2 nm. (c) HS-AFM image sequence of subsequent frames depicting CorA every 550 ms highlighting the structural flexibility of the molecule and the fast movements of the individual subunits. Full-frame size: 17.5 nm. Full z-scale: 2 nm. (d) HS-AFM image sequence depicting CorA every 250 ms. Full-frame size: 27 nm. Full z-scale: 2 nm.

concentrations. In saturating $Mg^{2+}$-concentrations (*Figure 5a*, green bars, *Video 7*), ~91% of the time molecules adopt the flower-shaped, 5-fold symmetric state, while the other states are barely populated. After $Mg^{2+}$-reduction to 3 mM (*Figure 5a*, blue bars) and full $Mg^{2+}$-depletion (*Figure 5a*, red bars, *Video 8*), the probability of finding CorA in any of the asymmetric states is reversed (*Figures 5a*, 3 classes on the right). Importantly, in 3 mM $Mg^{2+}$ ~50% of the molecules pool in the open-+ (others) class, representative of a highly mobile and flexible molecule that switches fast between numerous sub-structures that could not be classified to any of the conformations described by cryo-EM. After full depletion, re-addition of $Mg^{2+}$ could partially recover ~35% the symmetric closed-state structure (*Figure 5a*, yellow bars). Analysis of the probability of transitions from one structural state to the others (*Figure 5b*) demonstrated that under saturating $Mg^{2+}$-concentrations the symmetric closed state is, as expected, stable. Once CorA starts adopting any of the asymmetric conformational states in $Mg^{2+}$ concentrations ~3 mM and lower, the probability to switch back to the symmetric state is only ~14%. This probability is further reduced to ~7% at 0 mM $Mg^{2+}$.

Interestingly, comparing 3 mM and 0 mM $Mg^{2+}$-conditions gave a hint of the different $Mg^{2+}$-loads between the various asymmetric states: while the number of transitions between elongated states (*Figure 5b*, red outline: open-I and open-+) is virtually identical in these two conditions, the transition into a state with elevated height is strongly favored at 0 mM $Mg^{2+}$ (*Figure 5b*, dashed blue outline: open-II and open-I). We suggest that molecules with increased height where one sub-unit moves up close to the channel axis represent $Mg^{2+}$-free states.

## Discussion

On the basis of real-time HS-AFM imaging, we find that CorA $Mg^{2+}$-dependent gating can be best described in three phases: Above the apparent $Mg^{2+}$-affinity of the intracellular $Mg^{2+}$-sensor sites (~2 mM $Mg^{2+}$), the channel adopts a stable 5-fold symmetric state reminiscent of the high-resolution structures (phase 1). After several minutes of exposure to a $Mg^{2+}$-free solution, CorA transitions into a set of asymmetric architectures, which are characterized by elevated cytoplasmic

**Video 6.** High resolution HS-AFM movie of an individual CorA molecule switching between conformations in low $Mg^{2+}$ concentration. The molecule continuously switches between a subset of at least three states thought to represent conformations of the active CorA. Video settings: Full frame size: 17.5 nm, 35 pixels, scan rate: 2 frames $s^{-1}$, full color range: 2 nm.
https://elifesciences.org/articles/47322#video6

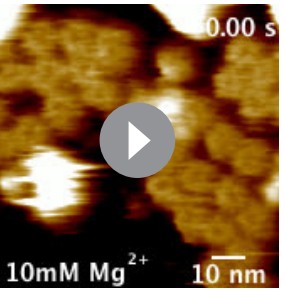

**Video 7.** CorA molecule cluster at saturating $Mg^{2+}$ concentrations. Molecules are found in the symmetric flower-shaped closed conformation more than 90% of the time. Video settings: Full frame size: 100 nm, 200 pixels, scan rate: 1.1 frames $s^{-1}$, full color range: 5 nm.
https://elifesciences.org/articles/47322#video7

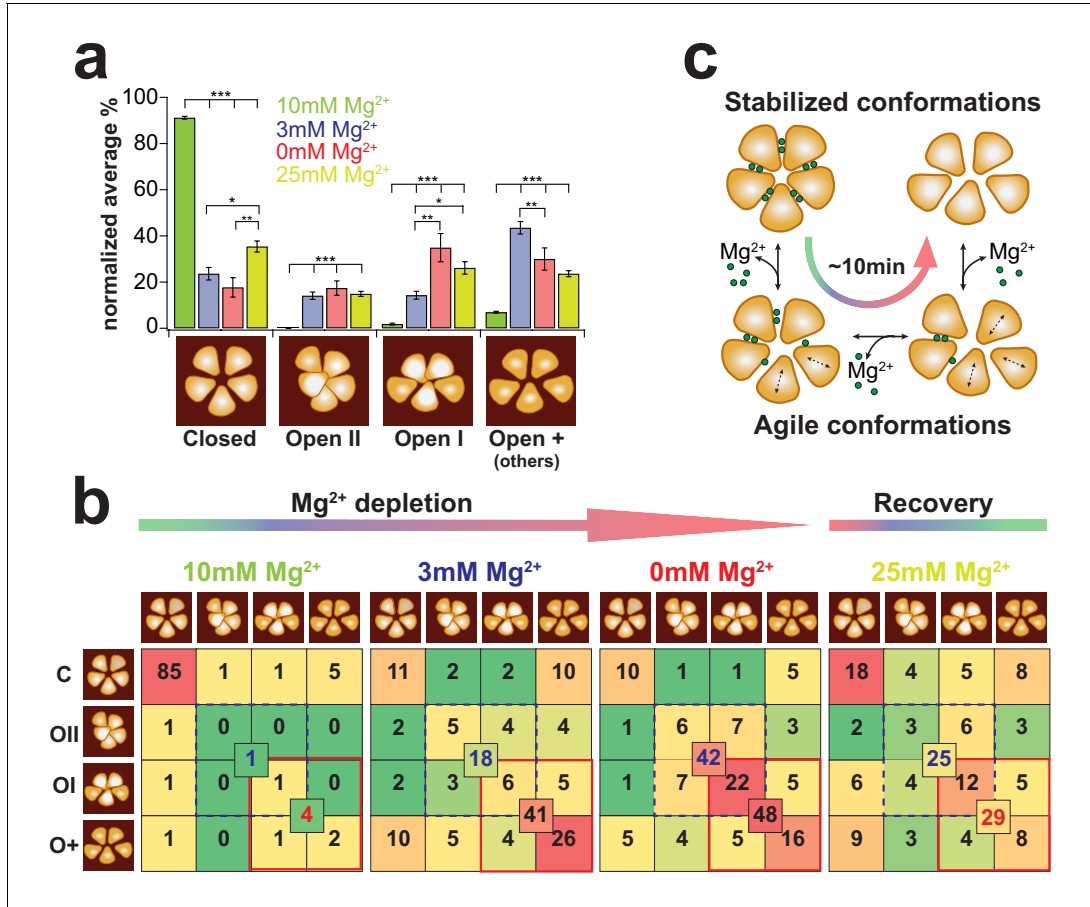

**Figure 5.** CorA state occupancy and transition dynamics. (**a**) State occurrence of 5-fold symmetric (assigned to Closed), dome-shaped (assigned to Open-II), elevated bean-shaped (assigned to Open-I) and other asymmetric CorAs (unassigned, Open +) at different $Mg^{2+}$-concentrations: 10 mM $Mg^{2+}$: green, 3 mM $Mg^{2+}$: blue, 0 mM $Mg^{2+}$: red, and after re-addition of 25 mM $Mg^{2+}$: yellow. Bars represent the normalized percentages of state assignments of ~20 CorA molecules in ~80–100 frames, ie up to ~2400 molecular representations, for each $Mg^{2+}$-condition. Error bars are standard error of mean (s.e.m.). Below, schematic representations of the various conformations. (**b**) CorA state transition-maps at 10 mM, 3 mM, 0 mM, $Mg^{2+}$ and after subsequent re-addition of $Mg^{2+}$ to 25 mM (from left to right). The schematic molecule on the left (rows) is the state in frame(n) and the schematic molecule on the top (columns) is the state in frame(n+1). Numbers are normalized percentages of the state transitions of the same experimental data as in (**a**). Color scale was adapted for each condition separately with a gradient from green (lowest occurrence of transition) over yellow and orange to red (highest occurrence of transition). Numbers in the center of boxes of 4 state transitions represent the sum of transitions between states with elevated subunits (blue dashed square) and between transitions of strongly elongated structures (red square). (**c**) CorA conformational transition model based on the HS-AFM observations. Within ~10 min of $Mg^{2+}$-depletion, the 5-fold symmetric, fully $Mg^{2+}$-liganded CorA transit into dynamically fluctuating molecules with flexible subunits until their conformation stabilizes in a $Mg^{2+}$-free highly asymmetric structure with increased membrane protrusion height. Figure 5 - Information Supplement 1: Estimation of thermally activated TM1 helix motions We estimated the theoretical range of helical motion by considering that TM1 behaves like a flexible rod undergoing thermally excited motions. The helix (rod) is characterized by a specific persistence length $L_P$ that is related to the bending stiffness $K_S$ through $L_P = \frac{K_S}{k_B T}$. The basic description for the change in curvature between two points on the rod is given by $\frac{\partial \vec{t}(s)}{\partial s}$, with s being the arc length and $\vec{t}$ a unit tangent vector at position (s). In an ideal system, the total elastic energy $E_{ela}$ of a particular conformation is given by the integral of the bending energies accumulated along a rod with contour length L: $E_{ela} = \int_0^L \frac{K_s}{2} \left(\frac{\partial \vec{t}(s)}{\partial s}\right)^2 ds$ Assuming only circular curvatures along the rod, $\frac{\partial \vec{t}(s)}{\partial s} = \frac{1}{r}$, where r is the radius of curvature. Using this basic description of an elastic polymer rod and considering the persistence length of protein α-helices $L_p$ = 100 nm (as described in ***Choe and Sun, 2005***) and a contour length L = 11 nm (length of TM1, see Supplementary Figure 4), we obtain $E_{ela} = \frac{L_P k_B T L}{2 r^2}$ This equation thus estimates that the radius of curvature r = ~23 nm at 1 $k_B$T. Helix bending of that range would result in ~2 nm movements at its end.

The online version of this article includes the following figure supplement(s) for figure 5:

**Figure supplement 1.** TM1 architecture and sequence conservation.

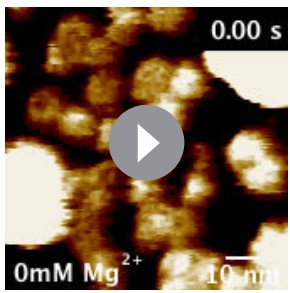

**Video 8.** CorA molecules monitored just after complete Mg²⁺-depletion. The majority of the molecules are in a highly dynamic state constantly switching between different structural states. Video settings: Full frame size: 100 nm, 200 pixels, scan rate: 1.8 frames s⁻¹, full color range: 5 nm.
https://elifesciences.org/articles/47322#video8

domains (phase 3). These structures are adopted through major conformational changes implicating hinge-bending of TM1 and reorientation of the cytoplasmic domains. This picture is wholly consistent with data derived from functional (electrophysiology), biochemical (crosslinking), biophysical (EPR, Fluorescence) and structural (cryo-EM) approaches. However, these results contrast with recent crystallographic data of CorA Mg²⁺ binding site mutants that reported limited or no structural changes compared to the closed wt channel, thereby challenging the hypothesis that Mg²⁺-depletion leads to large conformational changes and channel opening (*Kowatz and Maguire, 2019*). We argue, that the constrains of the crystal lattice on CorA limits the range of conformational changes that can be observed in X-ray structures. In contrast, HS-AFM imaging under physiological conditions shown here is more consistent with the single particle cryo-EM data reported in *Cleverley et al. (2015)*; *Matthies et al. (2016)* and the EPR spectroscopic data (*Dalmas et al., 2010*; *Dalmas et al., 2014b*), that is when the Mg²⁺ is investigated under lattice-free conditions. These facts must be considered a critical factor in explaining these differences. Among multiple conformational states within phase 3, we identified CorA structures that resembled the low-resolution cryo-EM open-I and open-II structures. These phases 1 and 3 are characterized by structural stability or at least limitations by the structural freedom of CorA. Here, more work is required to identify and clearly distinguish between different sub-structures in the Mg²⁺-free elevated state of CorA. In contrast, at intermediate Mg²⁺-concentrations, and/or after short incubation times at 0 mM Mg²⁺, the channel reveals a highly mobile and fluctuating state (phase 2). Active subunits change and the entire molecule displays high structural variability, likely adopting many more conformations than the three states so far identified by cryo-EM. Indeed, while we can assign the fully 5-fold symmetric state with certainty, our assignment of molecules to state open-I and open-II must be considered as putative. Perhaps more importantly, at intermediate Mg²⁺-concentrations ~ 50% of the molecules could not be assigned and were pooled in a class of undefined quaternary structures (open-+). In this context, the details of the cryo-EM study become relevant: open-I and open-II only pooled 26,271 and 27,416 particles, respectively, of the total 173,653 particles, and thus a significant portion of the particles remained outside of these classes (*Matthies et al., 2016*). We note, however, that this initial assignment is ultimately constrained by the resolution limitations of both cryo-EM structural assignments and the present application of HS-AFM. We expect that given a larger number of particles to process (in cryo-EM) or improvements in the spatial and/or temporal resolution of HS-AFM, additional states are likely to emerge. We propose that the various CorA structural states and the associated subunit movements observed by HS-AFM may reflect variations of number and location of Mg²⁺-ions bound to the cytosolic domain (*Figure 5c*).

Among the main structural features of CorA is its long (~11 nm) first transmembrane helix (TM1), ranging all the way from the periplasmic face to the surface of the cytoplasmic Mg²⁺-sensor. TM1 is actually the only connection between the TM region and the cytoplasmic domain (*Figure 5—figure supplement 1*). Thus, it seems plausible that the large fluctuations of the cytoplasmic domains might be directly translated by TM1 into fluctuations within the channel pore. We suggest this mechanism as the basis for ion conductance through a particularly long (in comparison to other channels) ~55 Å pore, in which moving TM1s allow the asynchronous progression of ions through the pore. Considering the typical persistence length of an α-helix of ~100 nm (*Choe and Sun, 2005*), allowed us to estimate how much an 11 nm long thermally activated helix fluctuates (*Figure 5* – Information Supplement 1). We find that at $1k_B$T TM1 could bend at its cytoplasmic end ~2 nm out of axis, in good agreement with observed bending of this helix in the open-I and open-II cryo-EM structures and the HS-AFM movies of subunit displacement. Thus, we propose that in the open state the pore is by far less narrow than assumed based on static structures. Indeed, constantly fluctuating TM1s

would provide a considerable channel diameter towards the cytoplasmic side. This model would predict that amino acids close to the periplasmic side of the pore play the key role for gating. In agreement with this hypothesis, we find that the $Mg^{2+}$-channel signature motif (GMN) is located right at the periplasmic end of the channel, whereas amino acids towards the cytoplasmic face of TM1 are less conserved (*Figure 5—figure supplement 1*).

The idea that there are multiple conductive conformations of CorA, or, as a paradigm shift, the conductive conformation of CorA is a fluctuating molecule, is favored by both, the subunit movement analysis and state transition analysis. In the long-term absence of $Mg^{2+}$, CorA molecules adopt a rather stabilized asymmetric state resembling the putatively open state cryo-EM structures. However, a living cell under physiological conditions is unlikely to ever reach $Mg^{2+}$-concentrations below ~1 mM. Thus we hypothesize that *in cellula* the physiologically relevant open state is a collection of fluctuating conformational states. Thus, the conformational energy landscape of CorA gating would consist of two deep energy minima representing the stable closed (symmetric) and putatively open (asymmetric) conformations connected by a wide plateau in which a variety of open conformations are adopted through permanent molecular fluctuations. The compelling utility of HS-AFM for the study of macromolecular dynamics at high spatio-temporal resolution and in native-like conditions is demonstrated, elucidating a process so far inaccessible to other structural or biophysical techniques. The present data set and our proposed mechanistic interpretation of the conformational dynamics of $Mg^{2+}$-dependent CorA gating sets the stage for an unprecedented understanding of CorA as a 'reverse' polarity ligand gated ion channel with an unique gating mechanism.

# Materials and methods

## Protein purification

CorA from *T. maritima* was expressed and purified as previously described (*Dalmas et al., 2010*). Briefly, the CorA-Pet15b vector was used to transform, then express CorA in *E. coli* BL21 DE3. After cell harvesting and disruption, membranes were collected by ultracentrifugation and gently solubilized. The sample was then cleared by ultracentrifugation and purified using cobalt high affinity chromatography column (Clontech Laboratories). The concentrated protein sample (AMICON 100 kDa cutoff membrane filters, EMD Millipore) was homogenized by gel filtration (Superdex 200 10/300 GL column, GE Healthcare Bio Sciences) and equilibrated in 50 mM HEPES, pH 7.3, 200 mM NaCl, 20 mM $MgCl_2$, and 1 mM DDM).

## Protein reconstitution

For CorA reconstitution into liposomes, the protein was gently mixed with freshly prepared solubilized POPC-POPG (3:1) or DOPC-DOPE-DOPS (4:5:1) lipids (Avanti Polar Lipids) at low lipid to protein ratios (LPR) between 0.2–0.4 (w:w) at a total protein concentration of 1 mg/mL. After 4 hr equilibration, detergent was removed by addition of biobeads overnight.

## Sample preparation for HS-AFM

A 1.5 mm diameter muscovite mica sheet was glued on a HS-AFM glass rod sample support and mounted on a HS-AFM scanner. Reconstituted CorA membranes were adsorbed on freshly cleaved mica for ~5 min. Subsequently, the sample was rinsed with imaging buffer (50 mM MES, pH6.0, 200 mM NaCl) containing 10 mM $Mg^{2+}$.

## HS-AFM

All experiments were performed using HS-AFM (*Ando et al., 2001*) (SS-NEX, Research Institute of Biomolecule Metrology Co.) operated in amplitude modulation mode, using ultra-short cantilevers (8 µm) with a nominal spring constant of ~0.15 N/m and a resonance frequency of ~600 kHz in liquid (USC, NanoWorld). Videos of CorA membranes were recorded with imaging rates of ~1–2 frames $s^{-1}$ and at a resolution of 0.5 nm $pixel^{-1}$. The energy input by the AFM tip (estimated to ~1.5 $_kBT$, considering a 90% imaging amplitude of a 1 nm free amplitude)(*Miyagi et al., 2016*) was minimized by continuously adapting the drive and setpoint amplitude and optimizing the feedback parameters.

## Structural titration experiments

Monitoring the transition of CorA from $Mg^{2+}$-saturated to low/no $Mg^{2+}$-conditions was achieved by depleting $Mg^{2+}$ by either adding EDTA into the measuring solution or alternatively by buffer solution exchange to $Mg^{2+}$-free buffer using an integrated constant-pressure and constant-flow pump system (*Miyagi et al., 2016*). Experiments were performed with CorA membranes prepared from three different purifications and ~10 different reconstitutions. In total, about 50 transition experiments on different membranes patches were performed on different days (over ~30 days) using two different, but similar HS-AFM systems, and ~10 USC cantilevers. All recordings showed a clear structural transition from stable 5-fold symmetrical proteins to very dynamic molecules with increased height. About 20% of the recorded CorA transition movies were considered for further analysis.

## Data analysis

HS-AFM images were first-order flattened and contrast adjusted using laboratory-made routines in Igor Pro software (WaveMetrics). Videos were then drift corrected with respect to the membrane patch, or aligned on individual CorA molecules using an in-house developed analysis software plug-in for ImageJ (*Fechner et al., 2009*; *Husain et al., 2012*). Dimensions of the CorA were calculated by height histogram analysis (n = 20,500 height values) and cross section analysis (n = 25) for each condition. Estimation of dwell-times was based on CorA protrusion height: ΔHeight/time traces were generated by subtracting the minimum pixel value from the maximum pixel value in the $5 \times 5$ nm center area of the molecule. For the molecular height transition detection the ΔHeight traces were analyzed by a Step Transition and State Identification (STaSI) method in a MatLAB (MathWorks) routine (*Heath and Scheuring, 2018*; *Shuang et al., 2014*). StaSI indicated a minimum description length (MDL) for fitting the data with two ΔHeight states (where the increased height states comprises all conformational states with elevated topography). For in depth analysis, 25 molecules of two individual experiments were tracked over 900 and 1600 frames at different $Mg^{2+}$ conditions, resulting in ~30,500 analyzed frames. Molecules in different height states and the corresponding dwell-times were binned either over 20 events along the time axis. Note, the transition analysis only discerned between states where subunits fluctuate between different height levels and did not classify between sub-states that exposed equivalent height levels to the putative closed (symmetric) state or the putatively open, activated (asymmetric and elevated) state. Such, the height/time traces did also not discriminate which of the open states was assumed. Notably however, our experiments showed that structures exposing increased protrusion heights are likely representatives of $Mg^{2+}$-depleted molecules and thus detection of elevated height is a valuable fingerprint for the state transition. For conformational transition analysis membrane patches of ~20 CorA channels were imaged at $Mg^{2+}$ concentrations of 10 mM, 3 mM, 0 mM, and after re-addition of 25 mM. Extracted CorA molecules from high-resolution movie sections (1,000–2000 molecular representations extracted from about 20 molecules for each $Mg^{2+}$ condition) were manually assigned to different structural states of (C) symmetric flower-shaped, (O-1) asymmetric elevated, (O-2) dome-shaped, elevated and (O-+) all others. The significance of the occurrence of each conformation under the four tested $Mg^{2+}$ concentrations was tested using a two-tailed students test.

## Acknowledgements

The authors thank George Heath for providing the Matlab scripts for the STaSI trace analysis, and Yi-Chih Lin for valuable comments on data analysis.

## Additional information

### Funding

| Funder | Grant reference number | Author |
| --- | --- | --- |
| National Institutes of Health | R01GM120561 | Eduardo Perozo |
| National Institutes of Health | DP1AT010874 | Simon Scheuring |

The funders had no role in study design, data collection and interpretation, or the decision to submit the work for publication.

## Author contributions

Martina Rangl, Conceptualization, Data curation, Formal analysis, Validation, Investigation, Visualization, Methodology, Project administration; Nicolaus Schmandt, Formal analysis, Investigation; Eduardo Perozo, Conceptualization, Resources, Supervision, Funding acquisition; Simon Scheuring, Conceptualization, Resources, Supervision, Funding acquisition, Methodology, Project administration

## Author ORCIDs

Eduardo Perozo (iD) https://orcid.org/0000-0001-7132-2793
Simon Scheuring (iD) https://orcid.org/0000-0003-3534-069X

## Decision letter and Author response

Decision letter https://doi.org/10.7554/eLife.47322.sa1
Author response https://doi.org/10.7554/eLife.47322.sa2

# Additional files

## Supplementary files

• Transparent reporting form

## Data availability

All data generated or analyzed during this study are included in the manuscript and supporting files.

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
