## [Decision Letter]

**Acceptance summary:**

Mg^2+^bala is a critical element that is involved in various physiological processes. CorA is the bacterial ortholog of eukaryotic genes encoding a mitochondrial Mg^2+^ ion channel found in all eukaryotes. CorA function is negatively regulated by Mg^2+^. These channels are open when the binding site is unoccupied but they close in the presence of high Mg^2+^. Previous structural studies primarily using EPR and cryo-EM suggest that the channel undergoes a closed to open conformational switch in absence of Mg^2+^, which results in loss of five symmetry. Cryo-EM studies show that the open channel exists in at least two conformations which could either represent snapshots of a fluctuating channel or intermediates in the gating pathway. Recent crystallographic studies, however, suggest that the unbound channels may not undergo as much of a conformational change as suggested by the cryo-EM and EPR studies.

In this study, the authors have used high-speed AFM to monitor the time-dependent conformational changes as Mg^2+^ is removed from the channel. HS-AFM technique allows them to track conformational changes in a single CorA channel over time and correlate these changes with putative gating transitions. Functional experiments show that the gating process of the channel is relatively slow – currents decay over 15-20 minutes in electrophysiology experiments. Thus, the conformational changes are expected to be within the bandwidth of these measurements (500 ms per frame). Mg^2+^ depletion experiments show that the stable starting structure with five-fold symmetry becomes dynamic with time and ultimately settles to one or more asymmetric structures. One of the most interesting aspects of this study is that unlike most ion channels, the open channel conformation is both asymmetric and dynamic. Overall this is a technically work which advances the field both in terms of providing new information about the gating mechanism of CorA and demonstrating the power of HS-AFM to monitor single molecule structural transitions.

**Decision letter after peer review:**

Thank you for submitting your article "Real time dynamics of gating-related conformational changes in CorA" for consideration by *eLife*. Your article has been reviewed by three peer reviewers, including Baron Chanda as the Reviewing Editor and Reviewer #1, and the evaluation has been overseen by a Reviewing Editor and Richard Aldrich as the Senior Editor.

The reviewers have discussed the reviews with one another and the Reviewing Editor has drafted this decision to help you prepare a revised submission.

Essential revisions:

1) All of the reviewers are concerned about the two-state model assumption for data analysis. The authors themselves state that there are two open states and one closed state. The model fit shown in Figure 3A is clearly inadequate. The open conformation is quite heterogenous, correct? The kymograph shown really looks like 3 states to me (the third being at a lower height, near a Δ Height of 0). And if the Mg^2+^ bound state is the baseline, lowest-height state, why doesn't the "down" state have a Δ-Height of zero? The choice of 2 states affects the dwell time analysis (everything else in Figure 3), and without a better rationale, I don't see why this choice is valid. There are other possible reasons for poor fit. For instance, some of the conformational transitions in CorA is much faster than the sampling rate and this could result in aliasing. This is difficult to rule out but should be discussed.

Also the height histogram on the right side shows a very disperse distribution, only two not very distinct local maxima that were not taken as level thresholds (so why 2-state?). As this data depiction is crucial for all the analysis done in the rest of Figure 3 it needs to be clearly and convincingly communicated.

STaSI is a model-independent idealization algorithm but it tends to overfit data. Have the authors tried any of the other HHM based algorithms? Given that they are assuming a two-state model, it would make more sense to use HMM.

2) The reviewers are convinced that the CorA molecule becomes highly dynamic in the absence of Mg^2+^ and find it very interesting that this conformation is conducting. However, we all agree that the assignment of these highly dynamic structures to discrete states is restrictive and perhaps also subjective. It is not clear how the conformational states (closed, open I, open II) taken from EM work were assigned to the conformations recorded with high speed AFM. Was it done manually? What were the objective criteria? How many structures were taken? Figure 5B only gives percentages. Again, this is crucial for all the data analysis done in Figure 5. As the authors have stated, the classification of discrete state in this highly flexible molecule is a challenge. In Figure 5A, it is not clear whether the images shown below are idealized (expected) images or averaged images for each of the states. It should be stated clearly in the legend. The actual averaged image should be shown as an inset. In addition, in some instances, the state assignments are based on changes in the shape rather than height. If both heights and changes in shape are taken into consideration, it appears that the number of possible states will be much more and trying to bin these highly dynamic structures into a few discrete states becomes less meaningful.

3) My other main critique is that it is very difficult to understand how changes in Mg^2+^ concentration were achieved for each experiment, and why the experiments were designed the way that they were. It seems like two different methods were used to change Mg^2+^Mg^2+^ concentrations: either pipetting in EDTA, or slow perfusion with an EDTA-containing solution over 10's of minutes (Figure 2—figure supplement 2). (Subsection “Mg^2+^-depletion induces large conformational changes 1 of the intracellular face” paragraph two: it's unclear whether perfusion is performed with EDTA solution or not. This could be clarified in the main text so that the reader doesn't have to go to the supplemental figure legend).

I assume that adding EDTA nearly instantaneously removes bulk Mg^2+^, so that any slow kinetic component is due to channel kinetics. Whereas with perfusion, changes in Mg^2+^ concentration and the channel's conformational journey from "phase 1" to "phase 3" both occur on over a minutes-long time scale. Because of the slow kinetic component to channel dynamics, it's really important for the logic of the paper to be clear about which method of Mg^2+^ depletion was used in the data in Figures 2 through 5.

Do I understand correctly that many (all?) of the experiments in Figure 3 involved a ramp from high Mg^2+^ to low Mg^2+^ over about 14 minutes with another 12 minutes in Mg^2+^ free conditions (as shown in Figure 2—figure supplement 2)? In Figure 3, the sharp delineation of the green, blue, and red boxes seem to indicate Mg^2+^ concentration steps, and the legend refers to images collected at saturating, 2, and 0 mM Mg^2+^. This is confusing. The text goes back and forth between describing imaging at 2 mM (subsection “Mg^2+^-depletion induces large conformational changes 1 of the intracellular face” paragraph one and ~2 mM). According to my reading, Figure 5 then uses 3 discrete Mg^2+^ concentrations, but the use of the color-fade arrows seems to indicate a ramp.

I'm not clear what advantage ramping the Mg^2+^ concentrations provided. Given the amount of functional data already available for this channel, including a good idea of the Kd for Mg^2+^, I might have gone straight to imaging at discrete Mg^2+^ concentrations. Could the rationale be explained?

4) Abstract: "finally equilibrates to an asymmetric structure." Is the final state a single asymmetric structure or it is at least two conformations. Based on the state identification matrix in Figure 5, it would seem that at 0 Mg, there are more than one asymmetric structures, unlike the Mg bound state.

5) Abstract: – "putative open state adopts multiple conformations through hinge-bending motions". This sentence contradicts the previous sentence and it is not clear whether the data provided in this study show hinge-bending motion. The conformational changes are compatible with hinge-bending but there is no direct evidence here.

---

## [Author Response]

Essential revisions:1) All of the reviewers are concerned about the two-state model assumption for data analysis. The authors themselves state that there are two open states and one closed state. The model fit shown in Figure 3A is clearly inadequate. The open conformation is quite heterogenous, correct? The kymograph shown really looks like 3 states to me (the third being at a lower height, near a ΔHeight of 0). And if the Mg^2+^ bound state is the baseline, lowest-height state, why doesn't the "down" state have a Δ-Height of zero? The choice of 2 states affects the dwell time analysis (everything else in Figure 3), and without a better rationale, I don't see why this choice is valid.

CorA actually adopts many more than 2 structural states (indeed, that is the major point of this and previous manuscripts). As such, the analysis as presented in 3A is not based on explicit structural states nor determines dwell times directly associable with specific structural states. Let us first go over the analysis carried out in Figure 3 in some detail: Each molecule is analyzed as described in 3A and then the behaviors of all molecules are pooled in 3B and 3C.

Originally, we made a significant effort to track the movement of the individual subunits and make state assignments by transforming the CorA ring profiles into height cross-sections. Theses profiles were associated to the states described by cryo-EM. However, given the large variability within the molecules (lateral and rotational movements), combined to the temporal resolution limits of the experiment (and the bias from the polar transformation), it turned out to be impossible to determine states and/or individual subunit movements in any significant way.

Thus, as a subterfuge to define and assign a given structural state to each molecule at any instant, we simply tracked the height increase and fluctuations of CorA during Mg^2+^-depletion experiments. In light of the nature of the structural changes between the closed (Mg^2+^-bound) and many open (Mg^2+^-free) conformations, we are confident that height is a reliable measure to track the main closed-open transition. In other words, unable to define several states that could be integrated into a gating model as the reviewer suggests, we use a 2-state model with no other ambition than assigning OFF vs ON, where all of the poorly defined “open” state conformations are “pooled” as the ON state. This is clearly not ideal, but we argue that given the fact that the open state is unlikely to be defined in the classical, “single conformation” sense (as is one of the main conclusions of this work), this simple model is just intended to assess the kinetics of entry and exit of the closed state i.e. the symmetric state vs all other states.

How was this assessed: To determine CorA’s height changes we only used the center area of each molecule (5x5nm) and calculated the relative ΔHeight (the maximum pixel value minus the minimum pixel value). The reason that absolute height was not reported is mostly due to the larger variability in absolute height values under a wide number of experimental conditions. The region of interest of this measurement was defined in the center area of each molecule. In cases of limited resolution, the pore might simply not be observed in an otherwise low CorA molecule (ΔHeight <1nm). Hence, when considering straightforward changes in height, CorA formally switches between only 2 states, yet due to resolution limitations, the ΔHeight of the lower state is somewhat widened.

The major conformational difference between the closed and the multiple open states, leads to an increased height of the center or the rim by ~2nm ΔHeight in the open state(s) (e.g. one subunit ‘stands up’ in one of the bean-shaped asymmetric conformations). When all subunits are low but there is a central pore (e.g. closed state, 5-fold symmetric molecule) the observed height difference between pore and rim is ~1nm, and poorly resolved closed state molecules have a ΔHeight <1nm. In brief, failing to make significant structural state assignments, we used the best AFM criterion (i.e. height) to assign the closed state versus all other (putatively activated) states as a function of the progression of Mg^2+^-depletion. We thank the reviewers for pointing to these uncertainties and have changed the text accordingly as highlighted in the manuscript.

There are other possible reasons for poor fit. For instance, some of the conformational transitions in CorA is much faster than the sampling rate and this could result in aliasing. This is difficult to rule out but should be discussed.Also the height histogram on the right side shows a very disperse distribution, only two not very distinct local maxima that were not taken as level thresholds (so why 2-state?). As this data depiction is crucial for all the analysis done in the rest of Figure 3 it needs to be clearly and convincingly communicated.

The reviewer is correct. The fluctuations of the CorA subunits are indeed faster than our sampling rate of about 2 frames s^-1^. In Author response image 1, we show a video section of 2 CorA molecules with a doubled scan rate of 250ms per frame and detect structural differences in single frames (see 1.75s compared to neighboring images 1.5s and 2s). Certainly, the structural fluctuations are much faster than this, and likely the reason why our classification efforts failed. Given the bandwidth limitation of image acquisition and the dynamics of the inter-state transitions in CorA, we have concluded that it would be difficult to capture individual states (other than the Mg^2+^-bound state and Mg^2+^-free states) at the current speed of image acquisition. In consequence, the only reasonable measure is whether the molecule is in its Mg^2+^-bound symmetric state (that can be well resolved) or an overlap of Mg^2+^-depleted dynamic molecules. We discuss the aspect of scan speed, subunit dynamics and the ‘merging of all activated states into one state as defined by height’ in our revised manuscript version. As detailed above and in the response to question 1a), the HS-AFM analysis essentially reduces the data to a 2-state model (inactivated or activated).

**Author response image 1. respfig1:** HS-AFM image sequence of depicting CorA every 250ms.

STaSI is a model-independent idealization algorithm but it tends to overfit data. Have the authors tried any of the other HHM based algorithms? Given that they are assuming a two-state model, it would make more sense to use HMM.

In this point we disagree with the reviewer. We performed a thorough analysis how well STaSI performs, down to low signal to noise ratios (SNR=0.5), and found that STaSI makes the correct conservative 2-state assignment with ~95% accuracy. We compared STaSI with vbFRET (another Bayesian based approach for state assignment and state transition fitting) and StaSI performed better in our hands. We like that STaSI uses a minimum description length (MDL) and proposes the number of states in a fully unbiased way, resulting in idealized traces. How many structural states are hidden behind the assigned height-states is of course another question that would need time analysis of these height-states with large sample statistics. This is well beyond the scope of the present work.

2) The reviewers are convinced that the CorA molecule becomes highly dynamic in the absence of Mg^2+^ and find it very interesting that this conformation is conducting. However, we all agree that the assignment of these highly dynamic structures to discrete states is restrictive and perhaps also subjective. It is not clear how the conformational states (closed, open I, open II) taken from EM work were assigned to the conformations recorded with high speed AFM. Was it done manually? What were the objective criteria?

This is one of the key points we are trying to make. Here, we elaborate in more detail:

1) We agree with the reviewer that there is difficulty to assign snapshots of the highly dynamic molecule to the states observed by cryo-EM. In its dynamic (Mg^2+^-free) condition, CorA channels adopt a large variation of conformations and any attempt to automatically assign particular structures, was unsuccessful by lack of resolution, bandwidth and/or the fact that neighboring molecules touched the molecules under scrutiny (we also tried the EM-centered program Relion for particle classification).

2) We also argue that the cryo-EM structures (Open-I and Open-II) must represent conformations that sit in some sort of energy well (likely very shallow) otherwise, cryo-EM would not have been able to average any kind of structure (even at low resolution). However, the fact that cryo-EM was able to discriminate these two conformations by no means implies that there might not be many more. In fact, we are convinced this is the case and one important reason why it is so hard to make structural assignments under a regime of relatively low bandwidth.

3) Given 2), it is not unreasonable to argue that our data are somewhat equivalent to the cryo-EM open state low resolution structures.

4) How have we assigned molecules to individual classes? Manually. We used the following criteria (underlined) that correspond to features observable in the cryo-EM structures: “The channel adopts several structures that we classified into 4 categories: First, the fully-liganded channel corresponding to the 5-fold symmetric closed conformation. Second, a structure of reduced diameter with three asymmetrically distributed protrusions of increased height, assigned to the cryo-EM open-I state. Third, a round- or star-shaped conformation with an elevated center. This HS-AFM topography can be easily identified as it only displays small structural variations and resembles the cryo-EM open-II state.”

5) However, we came to the same conclusion as the reviewers that an explicit assignment to the cryo-EM open states was too restrictive. Hence, we introduced another class of “not assigned” molecules: “Finally, given that our tentative assignment to open-I and open-II states only covers a fraction of all molecular observations, a fourth category must necessarily include the remaining range of structural variations, which we named Open-+ states…”. Indeed, we see that the majority of the molecules are in this 4^th^ class.

6) Finally, a key conclusion – see Figure 5C – is that the channel goes from a stabilized Mg^2+^-bound to a more-or-less stabilized Mg^2+^-free state over a large range of dynamic conformations. Thus, we do not think that we were restrictive trying to ‘push’ our findings towards assignment with the cryo-EM structures.

How many structures were taken? Figure 5B only gives percentages. Again, this is crucial for all the data analysis done in Figure 5. As the authors have stated, the classification of discrete state in this highly flexible molecule is a challenge.

For each Mg^2+^-condition between ~1,000-2,000 molecular representations were considered for classification into the 4 subgroups and then normalized to the total number of frames. In particular 943 HS-AFM images of mostly stable closed CorAs were analyzed in saturating 10mM Mg^2+^-condition, 1,122 frames at 3mM Mg^2+^, 2,394 CorA snapshots for 0mM Mg^2+^, and 1,104 representations were classified for the recovery experiment after re-addition of 25mM Mg^2+^. These numbers are now mentioned in the Materials and methods section and the corresponding figure legend.

In Figure 5A, it is not clear whether the images shown below are idealized (expected) images or averaged images for each of the states. It should be stated clearly in the legend. The actual averaged image should be shown as an inset.

The CorA molecules shown in Figure 5A bottom and Figure 5B left and top are sketches depicting the main structural classes of CorA. To avoid confusion, we now clarify this it in the figure legend.

In addition, in some instances, the state assignments are based on changes in the shape rather than height. If both heights and changes in shape are taken into consideration, it appears that the number of possible states will be much more and trying to bin these highly dynamic structures into a few discrete states becomes less meaningful.

We agree with the reviewers and believe that there are many more structures in the dynamic open state of CorA, considering the observed structural variations in the presented HS-AFM videos (see our responses to comments above).

This is why we state: “Active subunits change and the entire molecule displays high structural variability, likely adopting many more conformations than the 3 states so far identified by cryo-EM. Indeed, while we can assign the fully 5-fold symmetric state with certainty, our assignment of molecules to state open-I and open-II must be considered as putative. Perhaps more importantly, at intermediate Mg^2+^-concentrations ~50% of the molecules could not be assigned and were pooled in a class of undefined quaternary structures (open +).”, and, “We expect that given a larger number of particles to process (in cryo-EM) or improvements in the spatial and/or temporal resolution of HS-AFM, additional states are likely to emerge.”

Due to the current limitations in high-resolution single particle HS-AFM data in this study (~5,600 molecular representations in total) more work is required to further optimize sample preparation, which will allow more observations, and HS-AFM technology to distinguish between possible different sub-structures in the open state. However, our experiments observed a clear Mg^2+^- and time- depended sequence of conformational events for CorA: First, a transition into highly mobile, elongated structures occurs, followed by a transition into features resembling to those of the Mg^2+^-free cryo-EM structures, these molecules have a clearly increased height. As detailed above, the height changes (around Figure 3) are used to automatically detect activation of the molecules only, while the specific shapes are assigned later (see response to 2a) to describe the transition in more detail. We have added additional text in the Discussion section to point out these limitations.

3) My other main critique is that it is very difficult to understand how changes in Mg^2+^ concentration were achieved for each experiment, and why the experiments were designed the way that they were. It seems like two different methods were used to change Mg^2+^ concentrations: either pipetting in EDTA, or slow perfusion with a EDTA-containing solution over 10's of minutes (Figure 2—figure supplement 32). (Subsection “Mg^2+^-depletion induces large conformational changes 1 of the intracellular face” paragraph two: it's unclear whether perfusion is performed with EDTA solution or not. This could be clarified in the main text so that the reader doesn't have to go to the supplemental figure legend).I assume that adding EDTA nearly instantaneously removes bulk Mg^2+^, so that any slow kinetic component is due to channel kinetics. Whereas with perfusion, changes in Mg^2+^ concentration and the channel's conformational journey from "phase 1" to "phase 3" both occur on over a minutes-long time scale. Because of the slow kinetic component to channel dynamics, it's really important for the logic of the paper to be clear about which method of Mg^2+^ depletion was used in the data in Figures 2 through 5.

Yes, the reviewer understood the two concentration change protocols and we have fully and explicitly described them now in the main text.

Do I understand correctly that many (all?) of the experiments in Figure 3 involved a ramp from high Mg^2+^ to low Mg^2+^ over about 14 minutes with another 12 minutes in Mg^2+^ free conditions (as shown in Figure 2—figure supplement 2)? In Figure 3, the sharp delineation of the green, blue, and red boxes seem to indicate Mg^2+^ concentration steps, and the legend refers to images collected at saturating, 2, and 0 mM Mg^2+^. This is confusing. The text goes back and forth between describing imaging at 2 mM (subsection “Mg^2+^-depletion induces large conformational changes 1 of the intracellular face” paragraph one and ~2 mM). According to my reading, Figure 5 then uses 3 discrete Mg++ concentrations, but the use of the color-fade arrows seems to indicate a ramp.

We thank the reviewer for pointing out the confusion of analyzed Mg^2+^-concentrations and the depletion methods used. We only used the fast and manual EDTA titration procedure to deplete Mg^2+^ in discrete steps. In all but 1 experiment (Figure 3A and 3B, bottom panel) shown in Figure 3, Mg^2+^ was fully depleted in a single titration step (i.e. addition of ~10mM EDTA). For further analysis (Figure 3C) only molecules monitored at 0mM Mg^2+^ were analyzed. For the in-depth analysis of the individual structural classes the distinct Mg^2+^ concentrations of 10mM, 3mM, 0mM and re-addition of Mg^2+^ to 25mM were achieved pipetting. We clarify the distinct Mg^2+^ concentrations in the text and all figure legends.

I'm not clear what advantage ramping the Mg^2+^ concentrations provided. Given the amount of functional data already available for this channel, including a good idea of the Kd for Mg^2+^, I might have gone straight to imaging at discrete Mg^2+^ concentrations. Could the rationale be explained?

We performed the ramping experiments using a high-precision buffer exchange system first, to test for the affinity of Mg^2+^ in our particular experimental set-up (reconstituted and densely packed CorA channels adsorbed on a sample surface), independent of previously determined apparent affinities. Concerning the Kd, the ramp experiments merely corroborate the findings from functional data. Anyway, ramps help us make sure that the addition of EDTA does not physically disturb and influence the measurements. We agree with the reviewer that the transition analysis itself should be performed at distinct Mg^2+^ concentrations, particularly when considering the long transition times. Accordingly, all CorA transition experiments used for in-depth analysis and presented in the main manuscript (Figure 2-5 and corresponding videos) were recorded and analyzed at distinct Mg^2+^ concentrations. We clarify this by rephrasing the main text in the revised manuscript and as discussed above (point 3).

4) Abstract: "finally equilibrates to an asymmetric structure." Is the final state a single asymmetric structure or it is at least two conformations. Based on the state identification matrix in Figure 5, it would seem that at 0 Mg, there are more than one asymmetric structures, unlike the Mg bound state.

We realize that the expression “equilibrates to an asymmetric structure” might be misleading. Our HS-AFM experiments revealed multiple, dynamically fluctuating asymmetric structures, that show an elevated Mg^2+^-sensor domain after spending ~10 minutes in Mg^2+^-free conditions. Due to the flexibility and the fast dynamics of the large and elevated substructures of CorA we are resolution limited, and thus can only make tentative assignments to the two found cryo-EM structures. Our high resolution CorA analysis at the onset of complete Mg^2+^-depletion (Figure 4 and 5) is highly suggestive of more than two open states of elevated heights. To this end, we changed the Abstract to “… that equilibrates to asymmetric structures of increased membrane protrusion height level.”

5) Abstract: "putative open state adopts multiple conformations through hinge-bending motions". This sentence contradicts the previous sentence and it is not clear whether the data provided in this study show hinge-bending motion. The conformational changes are compatible with hinge-bending but there is no direct evidence here.

Good catch. As mentioned in point 4, we corrected the misleading phrasing of a single asymmetric state (point 4 above) and have also clarified in the revised version of the manuscript that the observed CorA subunit movements in the putative open state are consistent with the hinge-bending movement detected by cryo-EM. As discussed in the text and Figure 5—figure supplement 1, TM1 is the only connection between the TM region and the cytoplasmic domain. Logically, the large rearrangements in the Mg^2+^ sensor are likely translated to the channel pore via hinge bending in TM1.